# Bing–Neel Syndrome in Waldenström Macroglobulinemia: Updates on Clinical Management and BTK Inhibitor Efficacy

**DOI:** 10.3390/cancers17203358

**Published:** 2025-10-17

**Authors:** Masuho Saburi, Naohiro Sekiguchi

**Affiliations:** 1Department of Hematology, Oita Prefectural Hospital, Oita 870-8511, Japan; masuho-saburi@oita-u.ac.jp; 2Hematology Division, National Hospital Organization Disaster Medical Center, Tokyo 190-0014, Japan

**Keywords:** Bing–Neel syndrome, Waldenström macroglobulinemia, lymphoplasmacytic lymphoma, MYD88, CXCR4, cerebrospinal fluid, MRI, ibrutinib, tirabrutinib, zanubrutinib, pirtobrutinib, chemotherapy, radiotherapy

## Abstract

Bing–Neel syndrome (BNS), a rare complication of Waldenström macroglobulinemia, is caused by the direct infiltration of lymphoplasmacytic cells into the central nervous system (CNS) without large-cell transformation. Since clinical manifestations are heterogeneous and may overlap with IgM-related neuropathies, BNS is often under-recognized and diagnosed late. Because of its rarity, no prospective studies on BNS have been reported so far. In 2025, a consensus panel from the 12th international workshop on WM updated the guidelines for BNS, recognizing Bruton’s Tyrosine Kinase Inhibitors (BTKis) as a standard therapy, clarifying imaging and cerebrospinal fluid assessments during follow-up, and introducing revised response categories. We synthesize current evidence on epidemiology, pathophysiology, and diagnostic work-up and propose practical algorithms to distinguish BNS from mimics. We also review conventional chemoimmunotherapy-based approaches and highlight emerging data supporting CNS-penetrant BTKis, such as ibrutinib, tirabrutinib, and zanubrutinib.

## 1. Introduction

Waldenström macroglobulinemia (WM) is a rare, indolent B-cell neoplasm characterized by bone marrow infiltration by lymphoplasmacytic lymphoma and the production of IgM paraprotein. Bing–Neel syndrome (BNS) denotes the direct infiltration of these malignant cells into the central nervous system (CNS). Although rare (occurring in approximately 1% of WM patients), it may present at diagnosis or later in the disease course and may occur independently of systemic progression [1,2,3,4,5,6]. Clinical manifestations are heterogeneous and often non-specific, contributing to diagnostic delay. Therefore, a structured work-up is recommended, with contrast-enhanced brain/spine MRI, CSF cytology/flow cytometry, and molecular assays [1,2,3,5]. In 2025, a consensus panel from the 12th international workshop on WM (IWWM-12) updated guidelines for BNS, recognizing zanubrutinib as a standard therapy, clarifying imaging and CSF assessments during follow-up, and introducing revised response categories, including Clinical Complete Response and Progressive Disease [1]. Retrospective cohorts reported high response rates with covalent Bruton tyrosine kinase (BTK) inhibitors. After failure with covalent BTK inhibitors, non-covalent BTK inhibitors provide a rational salvage approach with encouraging activity [7]. Guided by these developments, this review aims to summarize current concepts in the biology and clinical spectrum of BNS, to outline a diagnostic framework for BNS, and to synthesize recent therapeutic advances with an emphasis on BTK inhibitor-based strategies. As this is a narrative review, a focused literature search was performed in PubMed using the keyword “Bing–Neel syndrome” for the time frame of Jan/1995–Sep/2025. In addition, relevant conference abstracts and key articles not indexed in PubMed were manually reviewed and included when appropriate.

## 2. Epidemiology and Pathobiology

The incidence of WM is approximately 3 per million persons per year in the United States and Europe [6]. BNS represents a rare complication of WM, occurring in approximately 1% of real-world clinical cohorts [8,9] and about 1.5% in population-based data, with a 15-year cumulative incidence of nearly 2.6% [8]. The interval from WM diagnosis to BNS highly varies, with reported median times of approximately 3 years in multi-institutional series [8,9] and 8.9 years in national cohorts [10]. BNS may present at WM onset (15–36%) and arise without concurrent systemic progression [8,9]. On neuroimaging, abnormalities are detected in approximately 80% of patients, while a minority have normal MRI despite CSF confirmation, underscoring the need to integrate MRI with CSF studies [8,9]. CNS involvement may be leptomeningeal, parenchymal, spinal, or mixed [9,10,11]. Molecularly, BNS reflects the WM biology. MYD88 L265P is detectable in paired marrow and CSF/brain in the majority of evaluable cases, and allele-specific PCR markedly increases CSF diagnostic yield, with 29 of 30 patients testing positive in a recent UK cohort. Importantly, MYD88 L265P is also observed in most cases of diffuse large B-cell lymphoma of the CNS and is therefore not disease-specific. More broadly, in WM, recurrent genomic lesions include MYD88 L265P in >90% of patients, CXCR4 mutations in approximately 25–30%, and TP53 abnormalities in 8–12%, although the latter two have been associated with an adverse prognosis and treatment resistance in WM. In the context of BNS, the contributions of additional lesions, such as CXCR4 or TP53, remain less well-defined [11,12,13,14]. Prognosis has improved with modern therapy. Historical cohorts reported a 3-year overall survival (OS) rate near 59% driven by the progression of BNS [8], whereas contemporary real-world data incorporating BTK inhibitors have shown an OS rate of approximately 93% at 2–4 years [11], aligning with the consensus panel from IWWM-12 that survival may approximate that of WM when effective therapies are applied [1].

## 3. Clinical Presentation and Differential Diagnosis

### 3.1. Neurological and Clinical Manifestations

BNS presents with a broad, site-dependent neurological spectrum and no single pathognomonic feature. Leptomeningeal involvement commonly produces headache, nausea, vomiting, cranial neuropathies, and visual disturbances, whereas parenchymal disease may manifest with cognitive decline, seizures, aphasia, impaired consciousness, or focal motor deficits. Spinal involvement can manifest with focal deficits or spinal localizing neurology such as cauda equina. Mixed patterns are not unusual [2]. BNS may emerge at any point in the WM course, including at first presentation, and may occur without concurrent systemic progression. Ocular involvement is uncommon at approximately 4–5% and includes optic nerve and orbital infiltration and discrete masses. A careful ophthalmologic assessment is recommended because ocular DLBCL or hyperviscosity-related changes may mimic BNS. Sensory symptoms are frequent but may be misattributed to peripheral neuropathy, and asymmetry or non-length-dependent features heighten suspicion for BNS [1].

### 3.2. Distinguishing BNS from IgM-Associated Neuropathies

In clinical patterns, BNS often produces asymmetric, multifocal, or clearly central signs, such as pyramidal or cerebellar involvement, while IgM-associated neuropathies, including anti-MAG neuropathy, typically present with symmetric distal sensory ataxia evolving over months to years and demyelinating features in nerve conduction studies. A subacute course with cranial nerve involvement or radicular cauda-equina features favors BNS, whereas chronic indolent sensory ataxia argues for IgM-associated neuropathies [2,3]. Ancillary testing also diverges, with anti-MAG antibodies and demyelinating conduction changes supporting an IgM-mediated neuropathy, while BNS is supported by CSF pleocytosis, elevated protein, the detection of lymphoplasmacytic cells via cytology or flow cytometry, and leptomeningeal enhancement on MRI [5]. In WM patients with atypical or non-length-dependent neuropathy, clinicians should consider the possibility of BNS involvement [3]. Hyperviscosity may produce headache and visual symptoms, and evaluating associations with serum IgM or viscosity and retinal examinations help distinguish hyperviscosity-mediated events from BNS, a recognized diagnostic pitfall.

### 3.3. Other Mimics

Beyond IgM neuropathies and hyperviscosity, conditions reported to mimic BNS include primary CNS lymphoma or histological transformation to DLBCL, infectious meningitis, therapy-related leukoencephalopathy, autoimmune/inflammatory disorders, and cerebrovascular events; clinicoradiological correlations and, when necessary, histological confirmation are essential to avoid misclassification [5].

## 4. Diagnostic Work-Up

### 4.1. Imaging

To exclude the mass effect or obstructive hydrocephalus and to avoid post-procedure artifacts, contrast-enhanced MRI of the brain and entire spine is performed before lumbar puncture using a protocol that includes FLAIR and T1-weighted sequences pre- and post-gadolinium. The most common abnormalities encountered are leptomeningeal or dural enhancement, while parenchymal mass-like lesions are less frequent. These findings correspond to the two main radiological patterns of BNS, a leptomeningeal form and tumoral/parenchymal form, which may coexist in some patients [3]. Importantly, normal MRI does not exclude BNS when clinical suspicion remains high [15], as demonstrated in a UK cohort in which approximately 15% of patients were MRI-negative but CSF-positive [11].

### 4.2. CSF Studies

A CSF evaluation needs to include opening pressure, a differential cell count, protein and glucose, cytology, and multiparameter flow cytometry to establish B-cell clonality. Although cytological examination may reveal atypical lymphocytes with a plasmacytic morphology, a diagnosis based solely on cytology needs to be avoided. Flow cytometry may identify lymphoplasmacytic cells with a characteristic WM immunophenotype that is positive for CD19, CD20, CD22, CD79a, CD79b, CD27, and CD52, with plasma cells expressing CD138 and IgM, while CD5, CD10, and CD23 are rarely expressed. The immunophenotypic profile in the CSF needs to match that in the bone marrow. To prevent false positives, peripheral blood contamination needs to be avoided [2,3].

The IgM index may assist in detecting intrathecal IgM synthesis, calculated as “(CSF IgM [mg/L]/serum IgM [g/L])/(CSF albumin [mg/L]/serum albumin [g/L])” when combined with CSF immunofixation, and an elevated index supports intrathecal or clonal IgM production; however, no validated diagnostic cut-off exists [1,16]. Although CSF protein electrophoresis and immunofixation may be useful [2], their availability is limited and their diagnostic value in BNS remains under investigation.

Molecular studies including IGH gene rearrangements and MYD88 L265P testing are recommended, and where feasible, correlating CSF clonality with bone marrow may strengthen diagnostic certainty [2]. In BNS, IGH rearrangements and MYD88 L265P have been detected in ~94 and ~100% of cases tested, respectively, using high-sensitivity PCR [17]. It is important to interpret IGH clonality results in conjunction with flow cytometry and MYD88 mutation testing, as oligoclonal patterns can occasionally be observed in mimickers. Since CSF samples in BNS often contain only small amounts of tumor DNA, highly sensitive methods are particularly important. An identical IGH sequence in CSF and bone marrow strongly supports the diagnosis [2]. However, MYD88 L265P alone is insufficient for a definitive diagnosis because it is also found in a large percentage of primary CNS lymphoma cases [18,19] and, less frequently, in other low-grade B-cell neoplasms, such as CLL or MZL, which rarely involve the CNS [20,21,22,23,24]. False negatives may occur with a low disease burden, and false positives may arise from bloody taps [2].

The prevalence of CXCR4 mutations in BNS remains unclear, but they were previously reported to occur in 25–30% of patients with WM [25,26]. In a multicenter study on zanubrutinib for BNS, the CXCR4 mutation status was assessed in seven patients, with a positive result in one (14%) [27]. The low detection frequency of BNS may reflect the limited sensitivity of conventional assays, such as Sanger sequencing, whereas allele-specific PCR or droplet digital PCR provides greater sensitivity for detecting these changes [28]. Current testing may underestimate the true prevalence of CXCR4 mutations in BNS and emphasizes the need for more advanced molecular approaches, including next-generation sequencing, which are expected to become available in clinical practice in the future.

### 4.3. Tissue Diagnosis

When a discrete enhancing CNS lesion is present and CSF is non-diagnostic or unsafe to obtain, stereotactic brain (or meningeal) biopsy is advised to confirm parenchymal BNS and exclude histological transformation. Immunohistochemistry typically shows an IgM-positive lymphoplasmacytic phenotype, while molecular studies, such as MYD88 L265P mutation analysis and IGH rearrangements, are recommended to establish clonal identity through comparisons with bone marrow. Consistent with the consensus panel from IWWM-12, the role of histopathology remains central; however, there is now a greater emphasis on integrating molecular diagnostics, including MYD88 and CXCR4 testing, as well as next-generation sequencing. It is important to note that in clinical practice, biopsy may be limited by patient age, comorbidities, or deep lesion location, and thus, molecular testing on CSF or other less invasive specimens may become valuable [1,2,3].

### 4.4. Laboratory Adjuncts

Serum IgM measurements remain essential, and when symptoms raise concern, serum viscosity needs to be checked in order to identify hyperviscosity-related neurological or visual issues that may mimic BNS. A parallel evaluation for alternative neuropathy causes (e.g., anti-MAG antibodies, diabetes, and vitamin B12 deficiency) is also advised. Bone marrow examination is recommended not only to define the systemic disease status but also to provide a molecular comparator through MYD88, CXCR4, or IGH rearrangement analyses of CSF or tissue studies, which is consistent with the consensus panel from IWWM-12. This integration of molecular correlations represents an important refinement in current practice [1,2,3].

### 4.5. Proposed Diagnostic Algorithm

The diagnostic algorithm is summarized in Figure 1. A high level of clinical suspicion for BNS needs to be exercised in WM patients presenting with atypical or focal deficits, rapidly progressive or non-length-dependent neuropathy, cranial nerve involvement, or central signs. Brain and whole-spine MRI with gadolinium is performed prior to lumbar puncture. If imaging is positive or suspicion remains high despite non-diagnostic MRI, lumbar puncture with cytology, flow cytometry, IGH clonality, and MYD88 is performed. A comprehensive multimodal diagnostic approach should be applied rather than relying on molecular findings alone. If CSF remains non-diagnostic while suspicion persists, biopsy of an accessible lesion is conducted. In parallel, IgM-mediated neuropathies and other mimics are evaluated and hyperviscosity is treated when present [1,2,3,11]. There is no recommendation of a therapeutic diagnostic approach. Thus, it is considered important to monitor clinical symptoms closely, with short-interval follow-up and repeated diagnostic evaluations, until the BNS diagnosis can be firmly established.

## 5. Treatment Landscape

### 5.1. General Principles and Goals

Treatment should be initiated only in patients with symptomatic BNS [1]. Asymptomatic patients may be observed without initial treatment [2,3]. Goals prioritize neurological recovery and durable disease control, while complete remission is not mandatory. The choice of therapy needs to consider the disease pattern (leptomeningeal vs. parenchymal mass), prior WM treatments, patient fitness and comorbidities, and the CNS penetration of candidate agents. The response assessment has evolved across guidelines (Table 1). In previous recommendations [2,3], response categories, such as a complete response (CR), partial response (PR), stable disease (SD), and progressive disease (PD), were proposed; however, the definitions were heterogeneous and often focused separately on clinical, radiological, or CSF findings. In contrast, the consensus panel from IWWM-12 [1] consolidated these into a standardized response framework that explicitly defines a clinical complete response (CCR), PR, SD, and PD. Notably, CCR requires the integration of clinical, MRI, and CSF findings, while the framework also acknowledges that malignant cells may persist in CSF during BTK inhibitor therapy despite clinical and radiographical benefits. In the follow-up, two important modifications were introduced in the IWWM-12 consensus panel recommendations [1]. First, routine serial MRI examinations are no longer recommended; instead, a single repeat MRI at approximately 3 to 6 months is advised to document treatment response, with subsequent imaging reserved for clinical indications. Second, routine repeat lumbar punctures during follow-up are not recommended in the current era of BTK inhibitors, mainly due to procedure-related bleeding risk and the potential for IgM rebound when therapy is withheld.

### 5.2. Conventional Chemotherapy and Radiotherapy

High-dose methotrexate (HD-MTX) with or without HD-cytarabine (Ara-C), including MTX/Ara-C combinations and primary CNS lymphoma-inspired regimens, such as R-MPV (rituximab, MTX, procarbazine, and vincristine), has historically been a mainstay for BNS [2,3]. In a multicenter retrospective study including 34 patients, HD-MTX-based regimens were used in approximately 40% of cases and Ara-C-containing regimens in about 25%. Overall, these treatments achieved an overall response rate of 66% (28% CR, 38% PR), with a 3-year OS of 59%. Myelosuppression and infections were the most common toxicities [9]. In the largest multicenter study (*n* = 44) [10], MTX-based therapy, often with Ara-C and/or intrathecal chemotherapy, achieved radiological or clinical improvement in 70% of patients, with median PFS of 26 months and a 5-year OS rate of 71%. These findings support the efficacy of CNS-penetrating agents such as HD-MTX and Ara-C in the frontline management of BNS. Intensive CNS-directed therapy, followed by autologous peripheral blood stem cell transplantation (ASCT), has achieved durable remissions in BNS, as shown in a French multicenter series of 14 patients who received BEAM (carmustine or bendamustine, etoposide, cytarabine, melphalan) in seven cases, total body irradiation with melphalan in one and thiotepa-based regimens in six (combined with busulfan, cyclophosphamide/melphalan, or carmustine), yielding a 3–5-year OS of 84% without treatment-related mortality [29]. Similarly, in a UK cohort of seven patients with extensive CNS involvement treated with intensive CNS-directed chemoimmunotherapy and ASCT consolidation using thiotepa or carmustine-containing conditioning, all achieved clinical and radiologic responses with minimal toxicity and 0% 100-day mortality, supporting ASCT as a safe and effective option for selected, chemosensitive BNS cases [30]. However, with the ongoing paradigm shift towards less invasive therapies, ASCT is no longer routinely recommended [1], and lower-intensity or targeted approaches are now generally preferred, particularly for older patients or those with comorbidities. Furthermore, the consensus panel from IWWM-12 no longer recommends HD-MTX-based regimens as the standard first-line therapy [1]. These regimens are now reserved for younger or fit patients with aggressive disease. Lower-intensity purine analog or alkylator-based regimens, such as fludarabine [31,32], cladribine [33,34], and bendamustine-rituximab [35], have been reported as fixed duration, largely outpatient options. Nonetheless, hematological toxicity, including neutropenia and infections, remains a concern, and these regimens are considered to be less durable than BTK inhibitors in most contemporary practices. It should be noted that, depending on national drug licensing and reimbursement policies, chemotherapy may be preferred over BTK inhibitors for treatment-naïve patients in some countries.

Intrathecal chemotherapy, typically with MTX, Ara-C, and/or rituximab, has been used in BNS to achieve transient cytoreduction, particularly in leptomeningeal-predominant disease, and may serve as a bridge to systemic therapy [9,36,37,38]. However, since durable disease control is rarely achieved with intrathecal therapy alone, the contemporary consensus panel from IWWM-12 emphasized that it is not recommended as monotherapy but rather positioned it as an adjunct to systemic treatment [1]. Focal radiotherapy remains an important tool for mass-like lesions or urgent local control and may serve as salvage therapy for refractory foci [39]. However, the risk of neurotoxicity increases with broader fields, such as craniospinal irradiation, and durable remission with radiotherapy alone is rare, particularly in patients with systemic involvement. Contemporary guidelines emphasize conformal, site-directed dosing integrated with systemic therapy, generally reserving radiotherapy for selected cases rather than routine monotherapy [1,2,3]. Cytotoxic chemotherapy and radiotherapy remain options for bulky or rapidly progressive disease, but are now reserved for selective use, often combined with or sequenced after BTK inhibitors.

### 5.3. Covalent BTK Inhibitors

Covalent BTK inhibitors have markedly reshaped the therapeutic landscape of BNS. Table 2 summarizes retrospective studies reported on covalent BTK inhibitors for BNS. Their introduction was initially driven by case reports with the first-generation BTK inhibitor ibrutinib, which penetrated the blood–brain barrier and achieved both neurological and radiological responses in patients with CNS involvement [40,41]. These early findings provided the rationale for BTK inhibition as a feasible treatment strategy in BNS, contrasting with historical reliance on cytotoxic chemotherapy and radiotherapy [2,3]. Among covalent BTK inhibitors, ibrutinib has the most mature dataset in BNS. In the multicenter retrospective study (*n* = 28), symptomatic and radiological improvement occurred in 85 and 83% of patients, respectively, as the best response, and 47% achieved CSF clearance. Within the first 3 months, 84% had neurological benefits and 57% had radiological improvement. Two-year event-free survival (EFS) and OS rates were 80 and 81%, respectively, and the 5-year survival rate from BNS diagnosis was 86%. Importantly, responses were observed at both 420 and 560 mg daily with no clear dose–response difference, and clinical benefits were evident regardless of prior BNS therapy. Adverse events were reported in 45% of patients, with grade 3–4 events including neutropenia, pneumonia, muscle cramps, bleeding, atrial fibrillation, and ventricular tachycardia, while grade 1–2 toxicities, such as diarrhea, fatigue, rash, bruising, and mild cytopenia, were more common. Some patients required treatment discontinuation due to adverse events, and deaths were attributed to disease progression, infection, or comorbidities [42]. Regarding the second-generation BTK inhibitor tirabrutinib, a recent multicenter retrospective study (evaluable *n* = 21; median follow-up, 30.9 months) reported an overall response rate of 100% with complete responses in 55.5%. The median time to the best response was 5 months, and 30-month EFS and OS rates were 90.5 and 90.2%, respectively. Adverse events occurred in 76% of patients, with grade ≥3 events in 33%, including neutropenia and infection, while rash (all grade 2) was the most common lower-grade toxicity [43]. Although currently approved only in Japan for WM [44], tirabrutinib is indicated for primary CNS lymphoma, where clinical studies have demonstrated high CNS penetration [45], further supporting its rationale in BNS. The second-generation BTK inhibitor zanubrutinib is endorsed as a standard therapy for BNS in the consensus panel from IWWM-12 [1]. In the multicenter retrospective study (*n* = 30; median follow-up 13 months), 92% of evaluable patients had clinical improvement, including complete resolution in 44%. Radiological improvement was documented in 85% of those re-imaged, and no BNS relapse occurred at the data cut-off. Any-grade adverse events occurred in 47% of patients, with grade ≥ 3 events in 20% [27]. A single-institution retrospective analysis of nine BNS patients treated with zanubrutinib, including treatment-naïve and ibrutinib-intolerant cases, showed neurological and radiological improvements. The median time to neurological improvement was 6.4 months, and treatment duration in the naïve subset was prolonged (median, 26.2 months), with no progression being reported [46]. Overall, these findings establish covalent BTK inhibitors as the cornerstone of modern BNS therapy. Agent selection needs to consider comorbidities, tolerability, and availability. This reflects a clear shift from earlier guidelines [2,3], in which BTK inhibitors were regarded only as emerging options, to their current recognition as central to BNS management [1].

### 5.4. Non-Covalent BTK Inhibitor

The non-covalent, reversible BTK inhibitor pirtobrutinib is highlighted by the consensus from IWWM-12 as an active option for WM after exposure to covalent BTK inhibitors, while noting that definitive CNS pharmacokinetic data in BNS are still being refined and warrant further study [1]. Multicenter correspondence reported three cases of BNS progressing on ibrutinib that subsequently achieved rapid clinical, radiological, and hematological responses with pirtobrutinib [7]. In clinical practice, pirtobrutinib needs to be considered as a salvage option for BNS under covalent BTK inhibitor resistance or intolerance [1,2,7].

### 5.5. Combining a BTK Inhibitor with Rituximab

It currently remains unclear whether the addition of rituximab to a BTK inhibitor improves the outcomes of BNS because no prospective BNS-specific trials have been conducted. In systemic WM, combinations may enhance hematological responses; however, many experts start with BTK inhibitor monotherapy for BNS and reserve the addition of rituximab or cytotoxic agents for selected scenarios, such as slow responders, CXCR4-mutated disease, or when systemic WM control is required, balancing infection and neurotoxicity risks [4]. The consensus panel from IWWM-12 also underscores that treatment needs to be individualized to disease distribution (leptomeningeal vs. mass-dominant) and patient factors, with BTK inhibitor monotherapy being an appropriate default for many symptomatic cases [1].

### 5.6. Supportive and Preventive Care

The use of antimicrobial prophylaxis proportional to regimen intensity and patient-specific risks is recommended and complemented with preventive strategies, such as vaccination and vigilance for early infectious complications to mitigate the immunosuppressive burden of therapy [3]. Hyperviscosity needs to be treated with plasma exchange when present, and the early initiation of rehabilitation for gait or cognitive deficits is recommended. BTK inhibitor class toxicities (atrial fibrillation, bleeding, and hypertension) and agent-specific events, such as rash or cytopenias, need to be monitored [1]. Since BTK inhibitor-associated bleeding and IgM rebound risks with treatment persist, routine repeat lumbar punctures are not recommended in clinically responding patients, and neurological function and imaging need to guide follow-ups instead, with CSF re-evaluation reserved for specific conditions, including treatment discontinuation, persistent deficits, suspected transformation, or infection [1,3].

## 6. Practical Treatment Algorithm

### 6.1. Initial Therapy

Because of its extreme rarity, there are no prospective studies on BNS. Therefore, treatment algorithms are based on retrospective studies. A treatment algorithm is summarized in Figure 2. While general treatment principles are outlined here, the choice of therapy may vary according to drug licensing and healthcare resources in individual countries. Covalent BTK inhibitors (ibrutinib, tirabrutinib, or zanubrutinib) are recommended as the preferred first-line therapy for most patients with BNS, regardless of the fitness status, given their CNS penetration and favorable efficacy-to-toxicity profile. In patients with intolerance of or a suboptimal response to covalent BTK inhibitors, switching within the covalent BTK inhibitors class is recommended. In fit patients with mass formation, HD-MTX-based induction (±rituximab) needs to be considered when rapid debulking is required, while acknowledging toxicity and logistical challenges. Transitioning to BTK inhibitors may help sustain disease control and improve tolerability. Radiotherapy is generally reserved for focal presentations where urgent local control is needed. In unfit or older patients, or those with leptomeningeal-predominant disease, covalent BTK inhibitors are recommended as initial therapy. Intrathecal agents may be used selectively as adjunctive cytoreduction in meningeal-only disease with severe symptoms or when systemic therapy cannot be initiated promptly, but are rarely adequate alone and do not delay CNS-penetrant systemic treatment. Early follow-ups will emphasize clinical improvement, supported by single MRI at 3–6 months. Routine serial MRI or repeat lumbar punctures are not advised in responding patients [1].

### 6.2. Relapsed/Refractory BNS

Treatment selection needs to incorporate the patient’s prior exposure history to covalent BTK inhibitors. After the failure of covalent BTK inhibitors, pirtobrutinib or HD-MTX-based chemotherapy is typically prioritized, whereas the initiation of therapy with a covalent BTK inhibitor remains the preferred approach for BTKi-naïve BNS and is often favored over intensive chemotherapy unless rapid cytoreduction is required [1,4].

### 6.3. Role of Genotypes

MYD88 L265P mutations provide biological support for BTK pathway dependence and are detected in the majority of WM and BNS cases. Although not an absolute requirement for treatment benefit, their presence generally supports BTK inhibitor sensitivity. In contrast, CXCR4 mutations, most often WHIM-like truncations, may delay or blunt responses to BTK inhibitors. Evidence regarding the clinical relevance of CXCR4 and other mutations, such as TP53, in BNS is extremely limited, and current knowledge is largely extrapolated from WM cohorts. Therefore, genotype-driven treatment modifications in BNS cannot be recommended at this stage [1,4].

## 7. Unmet Needs and Future Directions

The optimization of BTK inhibitor strategies remains a key challenge in BNS. Important unresolved issues include the optimal choice of first-line agent (ibrutinib, tirabrutinib, or zanubrutinib), treatment duration, criteria for de-escalation or discontinuation, and management strategies for patients who remain positive for CSF despite clinical and radiological responses. Due to the extreme rarity of BNS, adequately powered randomized or head-to-head prospective trials are not realistically achievable. Even multi-institutional cohorts struggle with limited case numbers and heterogeneous prior therapies. Accordingly, the generation of robust real-world registries has become an essential priority. These platforms need to systematically capture diagnostic latency, the genotype status (including MYD88 and CXCR4), treatment exposure (type, sequence, and duration of BTK inhibitors or chemotherapy), and longitudinal clinical endpoints. Importantly, the incorporation of patient-reported outcomes and neurocognitive assessments will address quality of life domains uniquely impacted by CNS diseases. Recent UK registry experience illustrates both the feasibility and challenges of multicenter data harmonization, underscoring the urgent need for internationally standardized endpoints and prospective registry structures that support pooled analyses across national cohorts [1,4,11].

Multiple CNS-relevant modalities currently under development for WM deserve evaluation in dedicated BNS cohorts, including BTK degraders [47], venetoclax with documented CSF penetration [48,49], CAR-T approaches active in primary and secondary CNS lymphoma [50,51], and IRAK-4 inhibition targeting MYD88/IRAK signaling with CNS penetration [52,53]. The activities of these novel agents in BNS need to be defined. Although available analyses will inevitably rely on small case reports or limited series, the systematic accumulation of data on CNS pharmacokinetics and drug penetration will be important when considering their clinical application to BNS.

## 8. Conclusions

BNS is a rare complication of WM that requires timely recognition and a structured diagnosis. MRI, CSF cytology/flow cytometry, and molecular testing, including IGH clonality and MYD88, are central to confirming BNS and distinguishing it from IgM-related neuropathies. Most patients carry MYD88 L265P, while CXCR4 variants, although common in WM, are less consistently detected in BNS, and their role remains unclear. BTK inhibitors have transformed the therapeutic landscape. Ibrutinib has been available the longest, tirabrutinib shows the highest response rates, and zanubrutinib is now endorsed by the consensus panel from IWWM-12 as a standard therapy. In the event of covalent BTK inhibitor failure, pirtobrutinib offers a promising salvage treatment option. Treatment needs to be individualized according to patient fitness, disease pattern, and genotype, with modern approaches achieving survival close to that of WM.

## Figures and Tables

**Figure 1 cancers-17-03358-f001:**
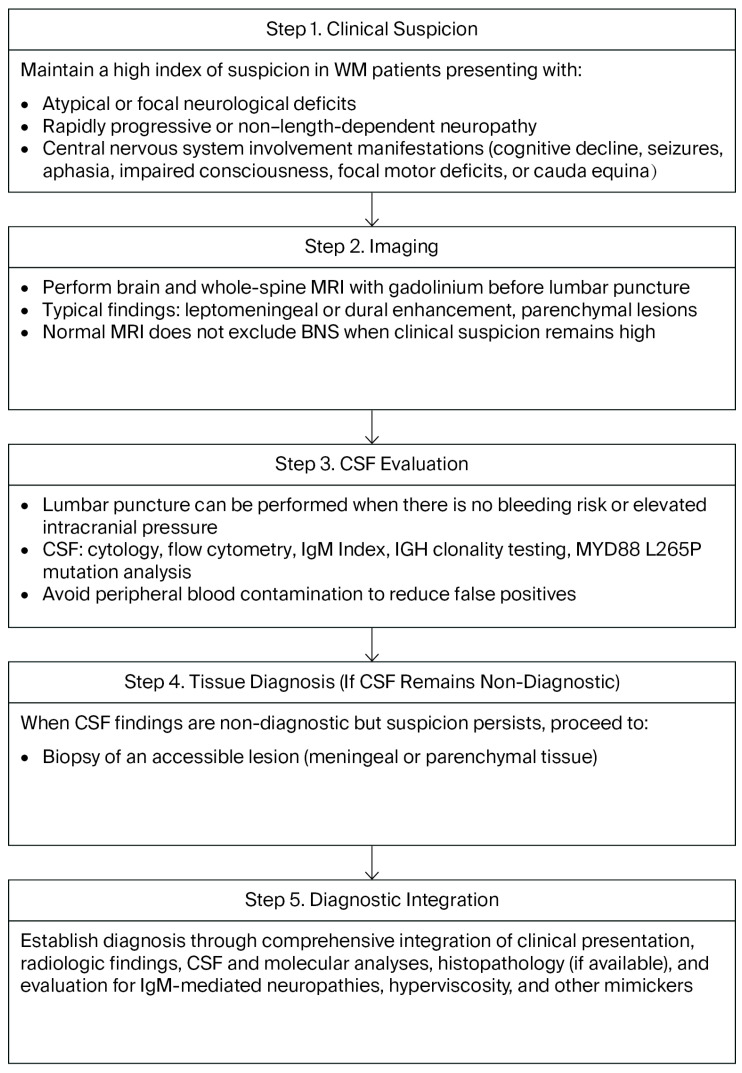
Proposed diagnostic algorithm for Bing–Neel syndrome. Stepwise approach integrating clinical suspicion, MRI, CSF studies, and histopathology. MRI with gadolinium should precede lumbar puncture; CSF analyses include cytology, flow cytometry, IgM index, IGH clonality, and MYD88 L265P testing. If CSF is non-diagnostic, tissue biopsy is recommended. Diagnosis requires integration of clinical, radiologic, CSF, and molecular data, while excluding IgM-mediated neuropathies and other mimickers. Abbreviations: BNS, Bing–Neel syndrome; BTKi, Bruton tyrosine kinase inhibitor; CSF, cerebrospinal fluid; HD-MTX, high-dose methotrexate; MRI, magnetic resonance imaging; IgM, immunoglobulin M; IGH, immunoglobulin heavy chain.

**Figure 2 cancers-17-03358-f002:**
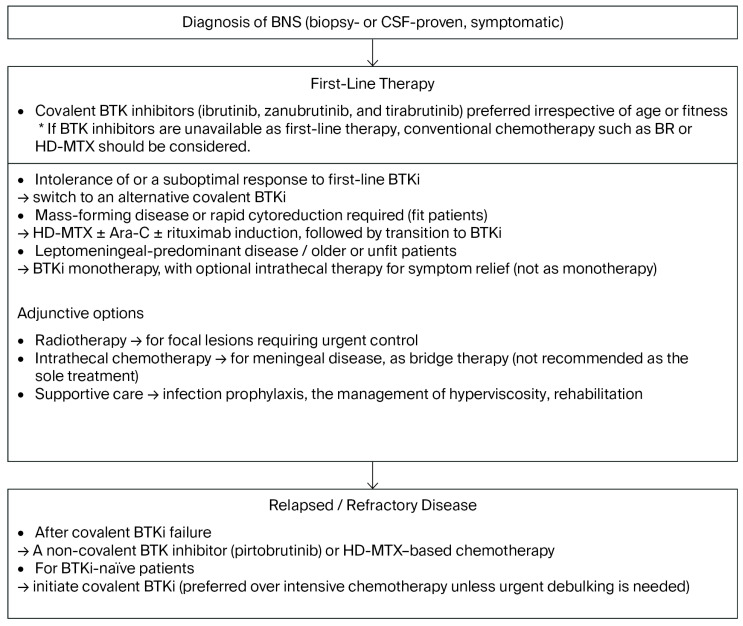
Proposed treatment algorithm for Bing–Neel syndrome. Notes (IWWM-12): Routine serial imaging is not recommended. Reserve CSF for selected conditions. BTKi may leave persistent CSF disease in ~50%. Avoid LP-related bleeding and IgM rebound from BTKi holds. The diagnosis of BNS requires biopsy or cerebrospinal fluid (CSF) confirmation in symptomatic patients. Covalent BTK inhibitors (ibrutinib, zanubrutinib, and tirabrutinib) are the preferred first-line therapy. Fit patients with mass-forming disease may start with HD-MTX ± Ara-C ± rituximab before transitioning to BTKi, while older, unfit, or leptomeningeal-predominant cases may receive BTKi monotherapy, with intrathecal therapy for symptom relief. Adjuncts include radiotherapy, intrathecal chemotherapy (not as monotherapy), and supportive care. In relapsed or refractory disease, pirtobrutinib or HD-MTX-based chemotherapy is recommended after covalent BTKi failure; covalent BTKi is preferred for BTKi-naïve patients. Genetic considerations include the presence of MYD88 mutations, which generally support BTKi sensitivity, while CXCR4 mutations may delay responses, although evidence remains limited. Follow-ups need to include one MRI 3–6 months after initial therapy to confirm responses, with subsequent imaging only if clinically indicated. Routine repeat lumbar punctures are not required for responding patients; however, a CSF reassessment may be warranted at treatment discontinuation in the presence of persistent neurological deficits or when transformation or infection is suspected. Abbreviations: Ara-C, cytarabine; BNS, Bing–Neel syndrome; BTKi, Bruton tyrosine kinase inhibitor; CSF, cerebrospinal fluid; HD-MTX, high-dose methotrexate; MRI, magnetic resonance imaging.

**Table 1 cancers-17-03358-t001:** Comparison of previously published response criteria (*Haematologica* 2017 [2]) and updated criteria proposed by the IWWM-12 consensus panel (2025) for Bing–Neel syndrome.

Response Category	Previous Criteria (*Haematologica* 2017) [2]	Updated Criteria (IWWM-12, 2025) [1]	Key Changes from Previous to Updated Criteria
Complete Response (CR)	Resolution of all neurological symptoms with the normalization of CSF and MRI findings	Resolution of all reversible neurological symptoms, with the normalization of CSF (cytology, flow cytometry, and MYD88 PCR) and MRI findings, and the absence of new neurological symptoms or MRI findings	Addition of MYD88 PCR in the CSF evaluation; explicit requirement of “no new symptoms/findings”
Clinical Complete Response (CCR)	Not defined	Resolution of all reversible neurological symptoms and MRI abnormalities attributed to BNS	New category introduced
Partial Response (PR)	Improvement in neurological symptoms, but with persistent radiological abnormalities and negative CSF	Improvement, but not complete resolution, of reversible neurological symptoms	Removed the requirement for negative CSF and persistent imaging abnormalities; simplified to a symptom-based definition
No Response (NR)	Persistence or progression of neurological symptoms, radiological findings, or CSF findings	No improvement in neurological symptoms related to BNS	Removed imaging/CSF requirements; defined solely on clinical symptoms
Progressive Disease (PD)	Defined only as “relapse”: reappearance of new signs/symptoms or progression/new MRI findings	Appearance of new or progressive neurological symptoms, or worsening of MRI findings attributed to BNS	Expanded from the relapse-only definition to include progressive disease

Table adapted from Minnema et al., *Haematologica* 2017 [2], and Sarosiek et al., *Semin Hematol* 2025 [1]. Abbreviations: BNS, Bing–Neel syndrome; CSF, cerebrospinal fluid; MRI, magnetic resonance imaging; PCR, polymerase chain reaction.

**Table 2 cancers-17-03358-t002:** Summary of BTK inhibitors in BNS.

	Ibrutinib (*Blood* 2019) [42]	Zanubrutinib (*Leukemia* 2025) [27]	Tirabrutinib (*Am J Hematol* 2025) [43]
Number of patients	28	30	21
Follow-up Duration	Median: 1.9 yrs from BNS diagnosis; 1.0 year from ibrutinib initiation	Median: 13 mo (range: 1–87 mo)	Median: 30.9 mo (range: 4.5–49.5 mo); from BNS diagnosis: 39.3 mo
Prior Therapies	Most had prior WM therapy; mix of chemoimmunotherapy and HD-MTX; some untreated at BNS diagnosis	67% prior WM therapy; 40% prior BNS therapy; 2 had prior ibrutinib	52.4% prior BNS therapy (IT chemo, HD-MTX, Ara-C, RT, F/C, BR); no prior BTKi
Time to Response	84% symptomatic, 57% radiological within 3 mo	92% of symptomatic patients improved within 3 mo	median time to best response: 5 mo
Response Rate	ORR 85% (CR 6%)	ORR 55% (CR 27%, PR 27%, NR 45%); clinical/radiological > 90%	ORR 100% (CR 55.5%)
Survival	2 yrs EFS: 80%; 2 yrs OS: 81%; 5 yrs OS: 86%	Median EFS not reached; OS not reached (no relapses observed)	30 mo EFS: 90.5%; 30 mo OS: 90.2%
Adverse Events	45% any AE; grade ≥ 3: pneumonia, arrhythmia, bleeding, neutropenia; 2 discontinuations	47% any AE; grade ≥ 3 in 20% (HTN, SCC, infection, FN); 3 discontinuations	76% any AE; grade ≥ 3 in 33% (neutropenia, lymphopenia, pneumonia, thrombocytopenia, appendicitis); 7 dose reductions, 8 interruptions, no discontinuations
Genetic Mutations (MYD88/CXCR4)	MYD88 L265P in 96% (CSF/BM); CXCR4 not reported	MYD88 L265P: 100% in CSF; CXCR4: 1/7 BM (14%) positive, 0/2 CSF tested	MYD88 L265P: 7/8 CSF (87.5%), 4/7 BM (57.1%); CXCR4 not tested

Abbreviations: AE, adverse event; Ara-C, cytarabine; BNS, Bing–Neel syndrome; BR, bendamustine plus rituximab; BTKi, Bruton tyrosine kinase inhibitor; CR, complete response; CSF, cerebrospinal fluid; dx, diagnosis; EFS, event-free survival; F/C, fludarabine plus cyclophosphamide; HD-MTX, high-dose methotrexate; HTN, hypertension; IT, intrathecal; mo, months; NR, no response; ORR, overall response rate; OS, overall survival; PO, per os (oral); PR, partial response; SCC, squamous cell carcinoma; WM, Waldenström macroglobulinemia; yrs, years.

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
