# Peer review of "Bing–Neel Syndrome in Waldenström Macroglobulinemia: Updates on Clinical Management and BTK Inhibitor Efficacy"

_cancers, 2025, doi:10.3390/cancers17203358_

Round 1

Reviewer 1 Report

Comments and Suggestions for Authors

This paper presents an interesting summary of a rare form of Waldenstrom's macroglobulinemia, Bing Neel syndrome. The authors propose a pratical treatment algorithm based on the results of published studies.

The presentation of a table with diagnosis algorithm could be interesting

Author Response

Comment

This paper presents an interesting summary of a rare form of Waldenstrom's macroglobulinemia, Bing Neel syndrome. The authors propose a practical treatment algorithm based on the results of published studies.

The presentation of a table with diagnosis algorithm could be interesting.

Answer

Thank you for this valuable comment. In accordance with this suggestion, we have added Figure 1, which summarizes a stepwise diagnostic algorithm for Bing–Neel syndrome, integrating MRI, CSF, and molecular testing. As a result, the previous treatment algorithm has been renumbered as Figure 2 in the revised manuscript.

Line 211: The diagnostic algorithm is summarized in Figure 1.

Line 410: Figure 1. Proposed diagnostic algorithm for Bing–Neel syndrome.

Stepwise approach integrating clinical suspicion, MRI, CSF studies, and histopathology. MRI with gadolinium should precede lumbar puncture; CSF analyses include cytology, flow cytometry, IgM index, IGH clonality, and MYD88 L265P testing. If CSF is non-diagnostic, tissue biopsy is recommended. Diagnosis requires integration of clinical, radiologic, CSF, and molecular data, while excluding IgM-mediated neuropathies and other mimickers.

Abbreviations: BNS, Bing–Neel syndrome; BTKi, Bruton tyrosine kinase inhibitor; CSF, cerebrospinal fluid; HD-MTX, high-dose methotrexate; MRI, magnetic resonance imaging; IgM, immunoglobulin M; IGH, immunoglobulin heavy chain.

Reviewer 2 Report

Comments and Suggestions for Authors

This is a comprehensive and timely review of Bing–Neel syndrome (BNS), an uncommon and clinically significant complication of Waldenström macroglobulinemia (WM). The manuscript is well-organized and synthesizes recent consensus updates. but some points to consider are:

1- this is a narrative review, but the literature search strategy is not well described.

2- The section on BTK inhibitors is extensively detailed, while non-BTK therapies (chemotherapy, radiotherapy, ASCT) are summarized more briefly.

3- the author stated imaging studies and CSF findings may not be typical of this disease, What is the best approach in theses cases? follow up?

4- since the mimickers may have oligoclonal band expansion which may not be easy to differentiate them from this entity. so rely only in IGH clonal studies may be misleading.

Author Response

Comment 1- this is a narrative review, but the literature search strategy is not well described.

Answer

We thank the reviewer for noting this point. Although this is a narrative review, we have now added a brief description of the literature search process in the Introduction section, clarifying the databases and main search terms used. Specifically, we performed a PubMed search using the keyword “Bing–Neel syndrome” for the time frame 1995/Jan–2025/Sep. In addition, important conference abstracts and key articles not indexed in PubMed were manually included to ensure comprehensive coverage. The following paragraph has been added to the Introduction section to describe this process.

Line 72: As this is a narrative review, a focused literature search was performed in PubMed using the keyword “Bing–Neel syndrome” for the time frame 1995/Jan–2025/Sep. In addition, relevant conference abstracts and key articles not indexed in PubMed were manually reviewed and included when appropriate.

Comment 2- The section on BTK inhibitors is extensively detailed, while non-BTK therapies (chemotherapy, radiotherapy, ASCT) are summarized more briefly.

Answer

Thank you for this valuable comment. We have expanded the descriptions of chemotherapy, radiotherapy, and autologous stem cell transplantation (ASCT) in the revised manuscript to provide a more balanced discussion.

Line 257: In a multicenter retrospective study including 34 patients, HD-MTX–based regimens were used in approximately 40% of cases and Ara-C–containing regimens in about 25%. Overall, these treatments achieved an overall response rate of 66% (28% CR, 38% PR), with a 3-year OS of 59%. Myelosuppression and infections were the most common toxicities [9].

Line 263: These findings support the efficacy of CNS-penetrating agents such as HD-MTX and Ara-C in the frontline management of BNS.

Line 265: In-tensive CNS-directed therapy followed by autologous peripheral blood stem cell trans-plantation (ASCT) has achieved durable remissions in BNS, as shown in a French multi-center series of 14 patients who received BEAM (carmustine or bendamustine, etoposide, cytarabine, melphalan) in seven cases, total body irradiation with melphalan in one, and thiotepa-based regimens in six (combined with busulfan, cyclophosphamide/melphalan, or carmustine), yielding a 3–5-year OS of 84% without treatment-related mortality [29]. Similarly, in a UK cohort of seven patients with extensive CNS involvement treated with intensive CNS-directed chemoimmunotherapy and ASCT consolidation using thiotepa or carmustine-containing conditioning, all achieved clinical and radiologic responses with minimal toxicity and 0% 100-day mortality, supporting ASCT as a safe and effective op-tion for selected, chemosensitive BNS cases [30].

Intensive CNS-directed therapy followed by autologous peripheral blood stem cell trans-plantation (ASCT), achieved sustained responses, including remission, for more than three years in eligible patients, with acceptable toxicity confirmed in small cohorts [29,30].

Comment 3- the author stated imaging studies and CSF findings may not be typical of this disease, What is the best approach in theses cases? follow up?

Answer

Thank you for this important comment. As described in Section 4.5 (“Proposed diagnostic algorithm”), in cases where imaging or CSF findings are non-diagnostic, further evaluation—including biopsy of an accessible lesion and assessment for IgM-mediated neuropathies and other mimickers—is recommended. We emphasize the importance of a comprehensive and integrated diagnostic approach to establish an accurate diagnosis in such diagnostically challenging cases. There is no recommendation for a therapeutic diagnostic approach. Thus, it is considered important to monitor clinical symptoms closely, with short-interval follow-up and repeated diagnostic evaluations, until the diagnosis of BNS can be firmly established. These points are added as follows.

Line 220: There is no recommendation of a therapeutic diagnostic approach. Thus, it is considered important to monitor clinical symptoms closely, with short-interval follow-up and repeated diagnostic evaluations, until the diagnosis of BNS can be firmly established.

Comment 4- since the mimickers may have oligoclonal band expansion which may not be easy to differentiate them from this entity. so rely only in IGH clonal studies may be misleading.

Answer

We agree with this important comment. A new sentence has been added emphasizing that IGH analysis should be interpreted in conjunction with flow cytometry and MYD88 testing, as oligoclonal patterns may occur in mimics. This clarification aims to highlight the need for a comprehensive multimodal diagnostic approach rather than relying on molecular results alone.

Line 170: It is important to interpret IGH clonality results in conjunction with flow cytometry and MYD88 mutation testing, as oligoclonal patterns can occasionally be observed in mimickers.

Line 217: A comprehensive multimodal diagnostic approach should be applied rather than relying on molecular findings alone.

Reviewer 3 Report

Comments and Suggestions for Authors

Abstract

line 26 suggest may overlap with IgM related 

line 28 Extreme rarity- can you provide incidence rates at this point?

line 35 - MyD88, please consider caveat that same mutation also often present in DLBCL of CNSL

Epidemiology

please provide baseline incidence of WM to frame the % incidence of BNS

line 79 - "CNS involvement" may also include spinal deposition, please add

Also, please consider adding caveat that MyD88 is not specific to WM as also found in 80-90% PCNSL

Clinical presentation

please consider adding where spinal disease can present with focal defects or spinal localising neurology eg cauda equine

Treatment landscape

line 220 -Can you reword as confusing - what about "asymptomatic" ?

Choice of therapy is also dictated by health economics/ licenced therapies by country

5.2 conventional chemo- consider noting around line 263, that treatment licencing may dictate utilisation of chemotherapy rather that BTKi in treatment naive patients  (eg in UK BTKi only licenced 2L+)

line 263 - suggest new paragraph from intrathecal, as it can be confusing with the length

Figure 1

Please consider in regions where BTKi are unavailable first line, conventional chemo such as BR (or R-MTX) should be considered

Author Response

Comment

Abstract

・line 26 suggest may overlap with IgM related

Answer

Thank you for this important comment. We have rephrased the sentence to “clinical manifestations may overlap with IgM-related neuropathies” in accordance with the reviewer’s suggestion.

Line 13: Since clinical manifestations are heterogeneous and may overlap with IgM-related neuropathies, BNS is often under-recognized and diagnosed late.

Line 26: Since clinical manifestations are heterogeneous and may overlap with IgM-related neuropathies, BNS is often under-recognized and diagnosed late.

Comment

・line 29 Extreme rarity- can you provide incidence rates at this point?

Answer

Thank you for this important comment. In accordance with the reviewer’s suggestion, we have added that the incidence of BNS has been reported as approximately 1% of patients with WM, prior to the sentence “Due to its extreme rarity, there are no prospective studies on BNS.” Accordingly, the phrase “Although the incidence of BNS is approximately 1% of WM” (Abstract, line 32) has been deleted.

Line 27: The incidence of BNS has been reported as approximately 1% of patients with WM. Due to its extreme rarity, there are no prospective studies on BNS.

Comment

・line 37 - MyD88, please consider caveat that same mutation also often present in DLBCL of CNSL

Answer

Thank you for this important comment. In accordance with the reviewer’s suggestion, we have added a note indicating that MYD88 L265P is also observed in most cases of diffuse large B-cell lymphoma of the central nervous system and therefore should not be considered disease-specific.

Line 37: Importantly, MYD88 L265P is also observed in most cases of diffuse large B-cell lymphoma of CNS and therefore is not disease-specific.

Comment

Epidemiology

・please provide baseline incidence of WM to frame the % incidence of BNS

Answer

Thank you for this important comment. In accordance with the reviewer’s suggestion, we have added information on the incidence of WM prior to the description of the incidence of BNS.

Line 77: The incidence of WM is approximately 3 per million persons per year in the United States and Europe [6]. BNS represents a rare complication of WM, occurring in approximately 1% of real-world clinical cohorts [8,9] and about 1.5% in population-based data, with a 15-year cumulative incidence of nearly 2.6% [8].

Comment

・line 86 - "CNS involvement" may also include spinal deposition, please add

Answer

Thank you for this important comment. In accordance with the reviewer’s suggestion, we have added potential spinal involvement as follows.

Line 86: CNS involvement may be leptomeningeal, parenchymal, spinal, or mixed [9-11].

Comment

・Also, please consider adding caveat that MyD88 is not specific to WM as also found in 80-90% PCNSL

Answer

Thank you for this important comment. The manuscript already notes in the diagnostic section that MYD88 L265P may also be detected in primary CNS DLBCL and therefore should not be regarded as specific to WM. Nevertheless, given the importance of this point, we have added further clarification in the indicated section that MYD88 L265P is not specific to WM, as it is also found in most cases of PCNSL.

Line 90: Importantly, MYD88 L265P is also observed in most cases of diffuse large B-cell lymphoma of CNS and therefore is not disease-specific.

Comment

Clinical presentation

・please consider adding where spinal disease can present with focal defects or spinal localising neurology eg cauda equine

Answer

Thank you for this important comment. We have added a sentence stating that spinal involvement can manifest with focal deficits or spinal localizing neurology such as cauda equina.

Line 108: Spinal involvement can manifest with focal deficits or spinal localizing neurology such as cauda equina.

Comment

Treatment landscape

・line 226 -Can you reword as confusing - what about "asymptomatic" ?

Answer

Thank you for this important comment. We have reworded the confusing sentence on line 226 for clarity, including a mention of asymptomatic or indolent cases as follows.

Line 226: Treatment should be initiated only in patients with symptomatic BNS [1]. Asymptomatic patients may be observed without initial treatment [2,3].

Treatment is indicated for symptomatic or biopsy- or CSF-proven BNS or when clinical combined with radiological evidence is compelling.

Comment

・Choice of therapy is also dictated by health economics/ licensed therapies by country

Answer

Thank you for this valuable comment. We agree that therapeutic choices are influenced by drug availability and national health economics. We have added a statement acknowledging that treatment options and access may vary by country depending on approved indications and healthcare systems and have incorporated this sentence into the revised manuscript.

Line 391: While general treatment principles are outlined here, the choice of therapy may vary according to drug licensing and healthcare resources in individual countries.

Comment

・5.2 conventional chemo- consider noting around line 253, that treatment licensing may dictate utilization of chemotherapy rather that BTKi in treatment naive patients  (e.g. in UK BTKi only licensed 2L+)

Answer

Thank you for this helpful suggestion. We agree that treatment availability and licensing can affect clinical practice. We have added a note indicating that, in some countries, chemotherapy may be used instead of BTK inhibitors in treatment-naïve patients due to national licensing restrictions and have incorporated this sentence into the revised manuscript.

Line 285: It should be noted that, depending on national drug licensing and reimbursement policies, chemotherapy may be preferred over BTK inhibitors for treatment-naïve patients in some countries.

Comment

・line 288 - suggest new paragraph from intrathecal, as it can be confusing with the length

Answer

Thank you for the suggestion. We agree that the paragraph was too long and could be confusing. We have created a new paragraph starting from “Intrathecal…” to improve clarity and readability. The intrathecal therapy section begins as a new paragraph for better readability.

Comment

Figure 1

・Please consider in regions where BTKi are unavailable first line, conventional chemo such as BR (or R-MTX) should be considered

Answer

Thank you for this helpful comment. We agree that, in regions where BTK inhibitors are not available as first-line therapy, conventional chemotherapy should be considered. We have added the following note to Figure 2: “If BTK inhibitors are unavailable as first-line therapy, conventional chemotherapy such as BR or HD-MTX should be considered.”

Line 390: The treatment algorithm is summarized in Figure 2.
